# Present-day correlations insufficient to predict cloud albedo change by anthropogenic aerosols in E3SM v2

Naser Mahfouz, Johannes Mülmenstädt, and Susannah Burrows

Pacific Northwest National Laboratory, Richland, WA, USA

**Correspondence:** naser.mahfouz@pnnl.gov

**Abstract.** Cloud albedo susceptibility to droplet number perturbation remains a source of uncertainty in understanding aerosol–cloud interactions, and thus climate states both past and present. Through E3SM v2 experiments, we probe the effects of competing processes on cloud albedo susceptibility of low-lying marine stratocumulus in the Northeast Pacific. In present-day conditions, we find that increasing precipitation suppression by aerosols increases cloud albedo susceptibility, whereas increasing cloud sedimentation decreases it. By constructing a hypothetical model configuration exhibiting negative susceptibility under all conditions, we conclude that cloud albedo change due to aerosol perturbation cannot be predicted by present-day co-variabilities in E3SM v2. As such, our null result herein challenges the assumption that present-day climate observations are sufficient to constrain past states, at least in the context of cloud albedo changes to aerosol perturbation.

## 1 Introduction

Marine stratocumulus clouds constitute a key source of uncertainty in aerosol–cloud interactions, which affect the Earth's radiation balance and thus climate projections (Intergovernmental Panel On Climate Change, 2023). One of the main challenges is understanding how these clouds respond and then adjust to aerosol perturbation. When aerosol concentrations are perturbed, their potential to act as cloud condensation nuclei is also perturbed, consequently perturbing cloud droplets resulting from said nuclei. The negative radiative forcing response to a cloud droplet perturbation is often decomposed into an instantaneous part followed by further parts due to adjustment of cloud properties, specifically water path and cloud fraction (e.g., Intergovernmental Panel On Climate Change, 2014; Bellouin et al., 2020).

The state-of-art understanding of processes involved in the radiative response due to cloud droplet perturbation is summarized by Zhang et al. (2022a). For thin, non-precipitating clouds at a constant liquid water path, an increase in cloud droplets yields higher concentration of smaller droplets, thus increasing cloud albedo (the Twomey effect; Twomey, 1977). For thicker clouds that are likely to precipitate, increasing cloud droplets can also lead to an increase in liquid water path, primarily through precipitation suppression induced by the increase in frequency of smaller droplets that are unlikely to precipitate, resulting in higher cloud albedo as well (the cloud lifetime effect; Albrecht, 1989). On the other hand, entrainment feedbacks resulting from cloud droplet evaporation (Ackerman et al., 2004) and sedimentation (Bretherton et al., 2007) tend to decrease liquid water path, resulting in reduced cloud albedo (Zhang and Feingold, 2023, and references therein). Both entrainment feedbacks are positive resulting from smaller droplets atop the clouds. For the evaporation–entrainment feedback, smaller droplets evaporate

faster, inducing more cooling and mixing, which in turn induces more droplet evaporation. For the sedimentation–entrainment feedback, smaller droplets decrease sedimentation flux atop the clouds, thus increasing cooling, which in turn increases the entrainment rate.

Using satellite observations, Zhang et al. (2022a) show that the albedo susceptibility of marine stratocumulus clouds in the Northeast Pacific Ocean can be diagnosed as a function of cloud state, revealing distinct regimes of radiative responses that can, in theory, be associated with different processes. In particular, their results reveal that the brightening Twomey effect is most dominant for thinner, non-precipitating clouds as well as the brightening cloud lifetime effect is most dominant for thicker, precipitating clouds; on the other hand, the darkening cloud-thinning processes are competitive in cloud regimes in between. Additionally, their diagnostic framework has been shown to be useful in analyzing different marine stratocumulus regions (Zhang and Feingold, 2023). Motivated by their findings, we adapt their methodology in analyzing climate model simulations to test if similar patterns emerge. We further design process-denial and process-scaling experiments to examine which, if any, processes can be detected by such diagnostics.

We note that atmospheric observations by satellites are limited to present-day conditions, whereas climate models allow us to explore past, present, and future scenarios. We therefore extend our analysis by performing simulations with pre-industrial aerosol emissions and precursors, keeping everything else constant at present-day levels. Said extension allows us to probe whether or not constraining cloud albedo susceptibility through present-day co-variabilities is informative for the anthropogenic cloud albedo change due to aerosol changes since the pre-industrial era. The anthropogenic albedo change is central to remaining uncertainties in aerosol–cloud interactions, and so is the central question of our study: Do present-day correlations predict anthropogenic cloud albedo change due to aerosols?

## 2   Methods

### 2.1   Modeling framework

We use the Energy Exascale Earth System Model (E3SM) v2 model in this study. The E3SM v2 components are documented and evaluated by Golaz et al. (2022). We briefly summarize the physics parameterization schemes most pertinent to cloud albedo susceptibility. The convection scheme is that of Zhang and McFarlane (1995) with some improvements in its trigger function (Golaz et al., 2022). The Cloud Layers Unified By Binomials (Golaz et al., 2002) is used for cloud macrophysics while the microphysics parameterization MG2 is that of Gettelman and Morrison (2015). The E3SM-modified four-mode Modal Aerosol Model (MAM4; Wang et al., 2020) is used with some further modifications as documented by Golaz et al. (2022).

In designing the experiments herein, we focus on two processes in MG2. The first is the sedimentation of cloud droplets. Cloud droplet sedimentation is calculated prognostically via the fall speed of droplets $v_{sed}$ (Morrison and Gettelman, 2008; Gettelman and Morrison, 2015). The empirical relationship $v_{sed} = aD^b$ (where $D$ is the droplet diameter, $a = 3 \times 10^7$ m$^{1-b}$ s$^{-1}$, and $b = 2$) is used to derive the mass and number fluxes in terms of the assumed droplet size distribution (Morrison and Gettelman, 2008, page 3647). We scale the fall speed of cloud droplets in our sensitivity cases by scaling the linear pre-factor

*a*. The second is the autoconversion of cloud droplets, following the parameterization of Khairoutdinov and Kogan (2000); the autoconversion rate scales with $\mathcal{A}N_c^\beta q_c^\gamma$ where $N_c$ is the in-cloud cloud droplet number concentration and $q_c$ is the liquid mixing ratio in the cloud (Morrison and Gettelman, 2008, page 3649). We adjust the droplet number exponent $\beta$ and linear pre-factor $\mathcal{A}$ simultaneously as we describe in the next section. (We do not adjust the liquid mixing ratio exponent $\gamma$ in this study.)

## 2.2 Simulation protocol

In this study, we strive to closely match the analysis of Zhang et al. (2022a) as well as Zhang and Feingold (2023) which is conducted on satellite data. To do so, we adjust the published configuration of E3SM v2 (Golaz et al., 2022) to facilitate closer comparison in terms of resolution. The first deviation from the published E3SM v2 configuration is in the form of resolution: We run our simulations at a dynamics and physics grid resolutions corresponding to approximately 28 and 42 km (ne120pg2; 0.25 and 0.38 degrees) as compared to the published 110 and 167 km (ne30pg2; 1 and 1.5 degrees). Thus, the effective resolution (28 km) is chosen to better match the Clouds and the Earth's Radiant Energy Systems (CERES) footprint resolution of 20 km. We note that a regionally refined configuration of E3SM v2 is documented by Tang et al. (2023) which has a fine grid resolution (like our protocol) around North America and coarse everywhere else (like the published v2 configuration), while maintaining a climate similar to the v2 configuration of Golaz et al. (2022). The second deviation is eliminating the minimum cloud droplet number limiter first introduced in E3SM v2 (10 cm$^{-3}$; Golaz et al., 2022) due to the centrality of cloud droplet number in our analyses.

Besides the above two deviations, all our experiments use the published E3SM v2 configurations documented by Golaz et al. (2022) and Tang et al. (2023), unless explicitly stated otherwise. Henceforward, we refer to the configuration with only the aforementioned deviations as the "default" configuration. To understand cloud albedo susceptibility in E3SM v2, we perform a variety of simulations and we highlight a select subset. We scale the cloud droplet sedimentation speed by 0, 1/4, 1 (default), and 4. We scale the autoconversion cloud droplet number exponent by 0, 1/2, 1 (default), and 2. When scaling the latter, we also re-balance the autoconversion linear pre-factor to ensure that residual radiation imbalance atop the model (RESTOM; roughly 1 W m$^{-2}$) and shortwave cloud radiative effect (SWCRE; roughly $-45$ W m$^{-2}$) are approximately the same across simulations. We conduct both sets of scaling experiments at present-day for all forcers and settings, that is, fixing sea ice, sea surface temperature, greenhouse gas concentrations, aerosol emissions and precursors, and land use at present-day conditions. For one of the seven experiments at present-day conditions, we repeat the runs with everything else the same except for pre-industrial emissions of aerosols and their precursors.

Finally, in order to improve the signal-to-noise ratio, we nudge our simulations to MERRA2 (Gelaro et al., 2017) for 15 months starting on October 1, 2010. The nudging protocol follows Zhang et al. (2022b) in nudging only the highest 70 of the 72 model layers to horizontal winds from the MERRA2 reanalysis dataset. The nudging is performed at a six-hourly cadence and has a relaxation timescale of six hours. Only the full 2011 calendar year is analyzed, allowing plenty of time (first three months) for spinning up the model in the sensitivity cases studied. Zhang et al. (2022a) as well as Zhang and Feingold (2023) use eight years of satellite data sampled once daily, whereas we use one year of data sampled eight times daily. Assuming that

half of our samples are at night time, we then have half of the data available in their analysis (since their snapshots are in the early afternoon local time). Nonetheless, the impact of data sampling has minor effect on the results (we verify this, but do not show it here, by using more years of data in our analysis for our default model configuration).

## 2.3 Analysis methods

For this study, we output diagnostic variables of interest instantaneously every three hours. In E3SM v2, cloud-top diagnostics can be calculated online following a modified version of the maximum–random cloud overlap assumption (Tiedtke et al., 1979, and references therein). We filter the data points satisfying conditions following Zhang and Feingold (2023). In particular, we select points satisfying solar zenith angle less than 65 ° or minimum solar insolation of 575 W m$^{-2}$, minimum cloud-top temperature 273 K, minimum cloud-top liquid cloud fraction 0.8, maximum cloud-top ice cloud fraction 0.2, and absence of convection trigger. We additionally limit our analysis to a predefined geographical region, the North East Pacific (15–35° N and 120–140° W), which is one of the key regions studied by Zhang and Feingold (2023) and the only region by Zhang et al. (2022a).

In order to construct the CDNC–LWP variable space presented by Zhang et al. (2022a), we convert grid-mean cloud-top cloud droplet number concentration, CDNC, and vertically integrated liquid water path, LWP, into their in-cloud counterparts using the cloud fraction. We infer the cloud albedo from the tautological relationship $A_{all} = f_c A_c + (1 - f_c) A_{clr}$ (Zhang et al., 2022a, equation 1) where $A$ is the albedo calculated as a ratio of reflected shortwave radiation to insolation and $f_c$ is the cloud fraction; the subscript c refers to cloudy-sky conditions, clr to clear-sky conditions, and all to both of them. We cluster the data points spatially (based on Euclidean distance) in 16-member groups. For each 16-member group at each time step, we calculate average values for CDNC, LWP, and cloud albedo susceptibility defined as the log–log regression of $A_c$ against CDNC. Finally, we present the figures as temporal averages inside CDNC and LWP bins as described in the figure captions.

## 3 Results

### 3.1 Default configuration

Following Zhang et al. (2022a) as well as Zhang and Feingold (2023), the main diagnostic result of our analysis is the cloud albedo susceptibility in the LWP–CDNC variable space. Zhang and Feingold (2023) hypothesize that said variable space delineates three distinct cloud albedo susceptibility regimes (cloud brightening and darkening regimes due to increasing droplet number) controlled by competing physics pathways. (We paraphrase the hypothesis of Zhang and Feingold (2023) for the rest of this paragraph.) First, for low LWP where the clouds are thinner and non-precipitating, the majority of cloud brightening could be attributed to the Twomey effect. Second, for high LWP and low CDNC where the clouds are thicker and likely to precipitate, cloud brightening or darkening could be attributed to the Twomey effect as well as the lifetime effect induced by precipitation suppression. Third, for the remaining areas, cloud-thinning processes (for example, entrainment feedbacks) likely dictate the cloud brightening or darkening regimes. Altogether, the hypothesis is: Cloud albedo susceptibility in the CDNC–

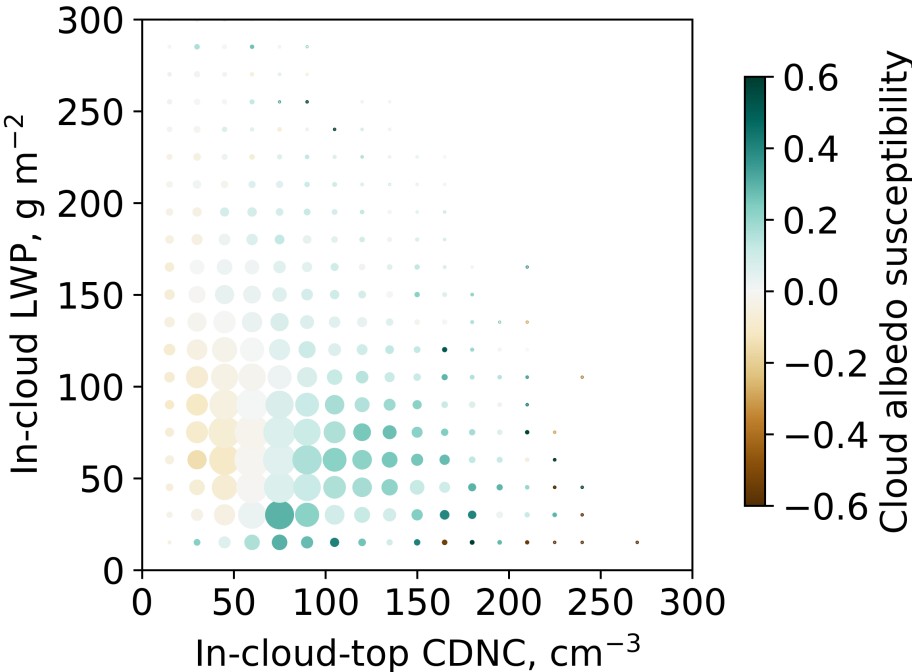

**Figure 1.** Cloud albedo susceptibility in the CDNC–LWP variable space for our default parametric configuration in present-day conditions. The cloud albedo susceptibility (color-scaled) is binned in both LWP (15 g m$^{-2}$ bins) and CDNC (15 cm$^{-3}$ bins). The size of the circles scales with the frequency of occurrence of each bin after filtering the data to highlight relative importance on the plot.

LWP variable space shows brightening due to Twomey and precipitation suppression effects throughout except for a region of
125 the space with high CDNC and high LWP where entrainment feedbacks dominate, resulting in cloud darkening.

Our default parametric configuration of E3SM v2 yields the results shown in Figure 1 in present-day conditions. Our results show the dominance of the Twomey effect for higher cloud droplet numbers. Yet, the competing effects of Twomey, precipitation suppression, and entrainment feedbacks cannot be delineated as cleanly as hypothesized in the preceding paragraph. In particular, our results show cloud darkening regimes when both CDNC and LWP are low (lower left quadrant in Figure 1).
Thus, our results diverge from those of Zhang and Feingold (2023) in that their darkening regime occurs at high CDNC and high LWP while ours at low CDNC and low LWP. We note that reconciling satellite and model studies is outside the scope of this work. There exist significant challenges in that regard (e.g., Ma et al., 2018; Quaas et al., 2020), and as such, we do not emphasize a direct comparison between our results and those of Zhang and Feingold (2023). We additionally note the E3SM v2 model exhibits a "too-frequent, too-light" precipitation bias among other biases in simulated cloud fields as studied elsewhere
(e.g., Xie et al., 2019; Zhang et al., 2024), and we do not quantify them here.

To better understand the underlying physics processes manifesting in the CDNC–LWP variable space, we conduct process-denial and process-scaling experiments described in the following section. We limit our attention to two processes: precipitation

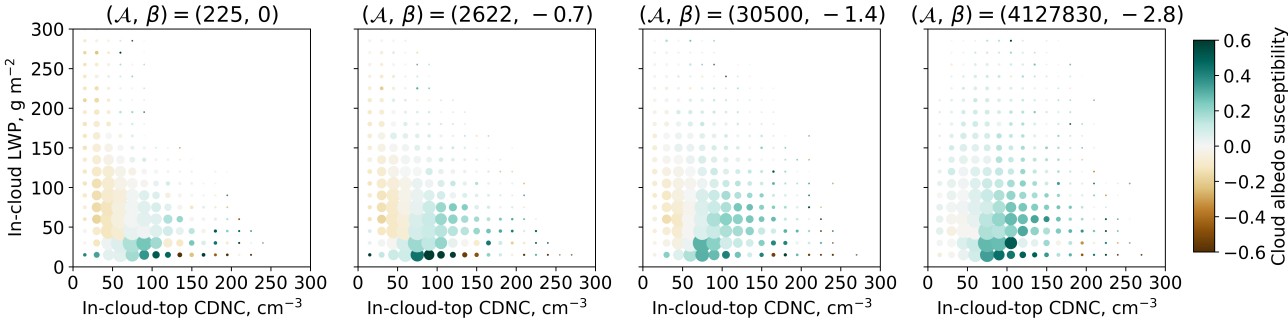

**Figure 2.** Like Figure 1, but for precipitation suppression scaling under default cloud sedimentation in present-day conditions. From left to right, the $(\mathcal{A}, \beta)$ pair is $(225, 0)$, $(2622, -0.7)$, $(30500, -1.4)$, and $(4127830, -2.8)$. The leftmost plot shows the "process-denial" experiment for precipitation suppression in the autoconversion parameterization. The remaining three are "process-scaling" experiments by halving and doubling $\beta$ around the default value of $-1.4$. The linear pre-factor $\mathcal{A}$ is modified to ensure the overall climatological state is loosely conserved between runs.

suppression via the autoconversion process and sedimentation-entrainment feedback via droplet fall speed. We acknowledge that both the satellite-based results by Zhang and Feingold (2023) and our model-based results represent two distinct representations of underlying physics that do not have to necessarily and exactly coincide, and thus our goal herein is to better understand which processes control cloud albedo susceptibility in the CDNC–LWP variable space in E3SM v2.

### 3.2 Process studies

In Figure 2, we vary the autoconversion parameters $\mathcal{A}$ and $\beta$. We present a precipitation-suppression process-denial experiment where we set the droplet number autoconversion exponent $\beta$ to zero while re-balancing linear pre-factor $\mathcal{A}$ to ensure roughly equivalent climatological settings (RESTOM and SWCRE at roughly 1 and $-45$ W m$^{-2}$). We additionally conduct two process-scaling experiments representing weakened and strengthened precipitation suppression in MG2. It is evident that precipitation suppression denial (leftmost panel) increases the prevalence and magnitude of cloud darkening due to increasing droplet number. Moreover, the precipitation suppression experiments show a gradual increase of cloud brightening as precipitation suppression is strengthened from left to right, culminating in mostly brightening clouds in the rightmost panel. In particular, cloud brightening due to precipitation suppression (rightmost panel) plays a role throughout the CDNC–LWP variable space, which is a departure from the hypothesized limit of precipitation suppression to the situations with low CDNC and high LWP (Zhang and Feingold, 2023). While said region (low CDNC, high LWP) lacks substantial data frequency in our model runs, it does show more brightening as the precipitation suppression is enhanced.

To further explore the underlying physics, we take the leftmost experiment in Figure 2, the experiment with precipitation suppression turned off. We take this specific experiment because we hypothesize that it allows us to see more clearly any effect of sedimentation–entrainment feedback in the absence of any competition from precipitation suppression. Around this particular experiment, we again conduct one process-denial and two process-scaling experiments, except this time targeting

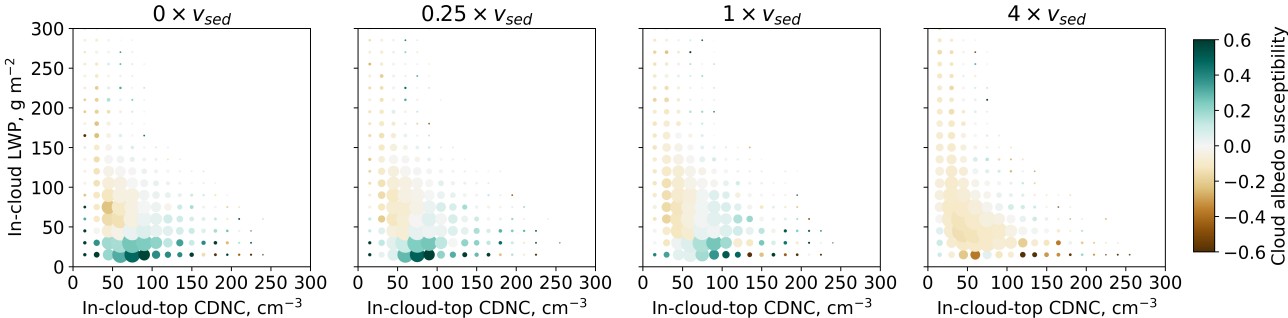

**Figure 3.** Like Figure 1, but for cloud sedimentation under no precipitation suppression in present-day conditions. We take the "process-denial" experiment from Figure 2 as the base line (third from left in Figure 2) and we change the cloud sedimentation via scaling the cloud droplet falling speed. From left to right, the cloud droplet falling speed is scaled by 0, 0.25, 1, and 4.

the cloud droplet sedimentation fall speed $v_{sed}$. In Figure 3, the effect of shutting both processes simultaneously shows slight cloud brightening (leftmost panel). On the other hand, the scaling of cloud droplet sedimentation fall speed shows an increase in cloud darkening from left to right, culminating in almost total cloud darkening in the rightmost panel. In the leftmost panel, where both precipitation suppression and droplet sedimentation are turned off, it is possible to see the brightening Twomey effect taking hold without any competition for thinner, non-precipitating clouds (low LWP).

In increasing the sedimentation fall speed by a factor of four, we set out to test whether or not we are able to enhance (and thus detect) the so-called sedimentation–entrainment feedback. In the original framing thereof, smaller droplets (resulting from a droplet number increase) evaporate faster and sediment less, increasing entrainment, and thus increasing the prevalence of smaller droplets — in a positive feedback loop (Bretherton et al., 2007; Zhang et al., 2022a). In Figure 3, we show that we are able to detect fingerprints of the sedimentation–entrainment feedback, though we are not certain about the specific pathways. We speculate that the size-dependent formulation of droplet sedimentation is the key mechanism through which increasing sedimentation causes an increase in the sedimentation–entrainment feedback. By construction, $v_{sed}$ is proportional to the square of the droplet diameter (Morrison and Gettelman, 2008, page 3647) and so when sedimentation is increased, there is a preferential sedimentation of larger droplets, leaving a distribution with more smaller droplets atop the cloud. We caution that definitively understanding the process(es) leading to the manifestation of this entrainment feedbacks in climate models is beyond the scope of this current manuscript. We note that recent work by Mülmenstädt et al. (2024a, b) provides insights related to diagnosing entrainment in general circulation models.

Overall, our results in Figure 2 and Figure 3 show that diagnosing cloud albedo susceptibility in the CDNC–LWP variable space (Zhang et al., 2022a; Zhang and Feingold, 2023) provides process-level understanding largely consistent with and explainable by our current understanding of physics involved. In Figure 2, manipulating the autoconversion parameter to increase precipitation suppression does in fact result in more brightening cloud regimes. In Figure 3, manipulating the size-dependent cloud droplet sedimentation fall speed to increase the sedimentation–entrainment feedback results in more darkening regimes. Together, both Figure 2 and Figure 3 bolster our confidence in the valuable insight provided by this method and its ability to

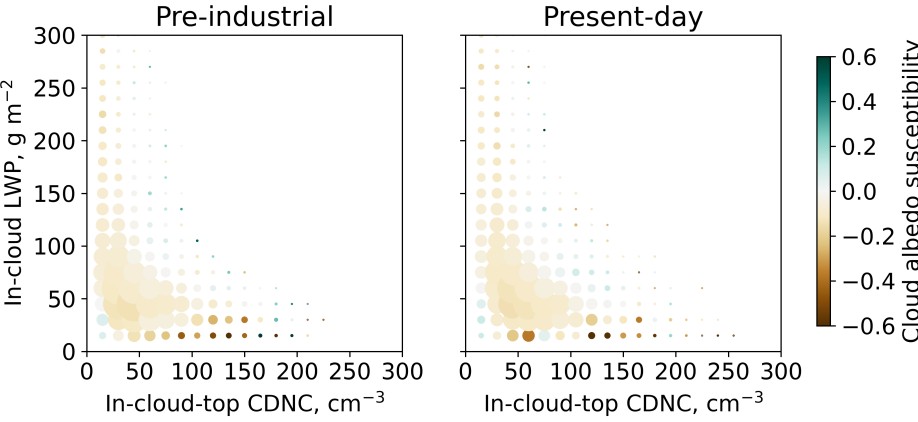

**Figure 4.** Like Figure 1, but for the configuration yielding darkening clouds everywhere, which is achieved by quadrupling cloud sedimentation under no precipitation suppression in pre-industrial (left) and present-day (right) settings.

diagnose the underlying physics, despite the imperfect alignment between results obtained from satellite observations (Zhang et al., 2022a; Zhang and Feingold, 2023) and model simulations (this work).

### 3.3 Pre-industrial–present-day comparison

In order to address the central question of our study, we arrive at Figure 4, which shows the configuration in the rightmost panel of Figure 3 at both pre-industrial and present-day conditions. Both results show darkening clouds (negative cloud albedo susceptibility) essentially everywhere based on present-day and pre-industrial co-variabilities between LWP and CDNC. By definition, a negative cloud albedo susceptibility means that cloud albedo change is negative under a positive CDNC change, and vice versa. As such, a reasonable interpretation of Figure 4 is: The cloud albedo will decrease as a result of an expected increase in CDNC going from pre-industrial to present-day emissions of aerosols and their precursors. But we find the opposite.

Because this is a climate model where we can simulate both pre-industrial and present-day aerosol conditions, we can rigorously assess if the perturbation in aerosol conditions yields the inferred change from variabilities. The mean CDNC in the pre-industrial case in Figure 4 is 38 cm$^{-3}$ while it is 49 cm$^{-3}$ for present-day conditions. This is a positive CDNC change. However, the mean cloud albedo in the pre-industrial case in Figure 4 is 0.363 while it is 0.377 for the present-day conditions. Therefore, despite the negative cloud albedo susceptibility in both present-day and pre-industrial settings, cloud albedo still increases as the CDNC increases, indicating positive cloud albedo susceptibility — contradicting the result obtained from correlations in both present-day and pre-industrial settings.

## 4 Conclusions

The main conclusion of this study is that the cloud albedo susceptibility inferred from present-day correlations in the E3SM v2 atmosphere model is insufficient to predict pre-industrial cloud albedo. This null result is important because it suggests that our current understanding of cloud albedo susceptibility may not be sufficient to predict how clouds will drive and respond to future climate changes. There are two caveats to our result. First, we only study a single climate model. Second, we do not yet have a definitive mechanistic understanding of why the model behaves the way it does. Nevertheless, our result is significant, as climate models are our primary tool for studying past and future climate states, and as highlighted by Mülmenstädt et al. (2024a), it challenges the underlying assumption that using present-day observations is sufficient to constrain past states.

Our sensitivity studies highlight two important details about E3SM v2 in the context of this work. First, it is possible to apply diagnostic techniques from satellite studies to climate model outputs by carefully designing climate model experiments. Second, the cloud regimes diagnosed in the model respond effectively to both the explicitly parameterized precipitation suppression and the implicitly parameterized entrainment feedback via sedimentation. Taken together, they bolster our confidence in the results manifesting due (and responding) to suspected physics pathways. However, we caution that more careful work is needed to better understand the mechanistic pathways, especially as our results do not match — in all their aspects — what is hypothesized in previous studies (e.g., Zhang et al., 2022a; Zhang and Feingold, 2023).

Overall, our study provides new insights into the limitations of our current understanding of cloud albedo susceptibility. We encourage further research to investigate the mechanistic pathways responsible for the behavior observed in E3SM v2 and to assess the implications of our results for other climate models and observational studies. While we focus on the present-day versus pre-industrial comparison in this study, we do not explicitly assess the implications for future scenarios, for example, for the purpose of better assessing cloud seeding proposals (e.g., marine cloud brightening) where the goal is to exploit cloud physics properties like cloud albedo susceptibility to cool the planet via a positive perturbation of cloud droplets via aerosols.

*Code and data availability.* The E3SM v2 model is used in this study, which is publicly available at https://github.com/E3SM-Project/E3SM (specifically commit 9dfef8b is used). The specific code version is additionally archived at https://doi.org/10.5281/zenodo.10436543 (Mahfouz, 2023b). All data outputs from the model runs analyzed in this manuscript are archived at https://portal.nersc.gov/archive/home/m/mahf708/www/casv2 and are documented at https://doi.org/10.5281/zenodo.10436618 (Mahfouz, 2023a). The raw data (approximately 3 terabytes in total) contain most two-dimensional model outputs related to aerosol–cloud interactions at an unstructured grid in two data streams: a monthly average and a three-hourly instantaneous snapshots. An end-to-end computationally efficient reproducer is provide and documented at https://zenodo.org/doi/10.5281/zenodo.10971987 (Mahfouz, 2024) which enables the full production of the first figure in this manuscript from archived raw data.

*Author contributions.* JM conceived the research idea herein and worked with SB to refine it. NM designed and conducted the experiments and analyses while consulting JM and SB throughout. NM wrote the manuscript with input from coauthors.

*Competing interests.* At least one of the (co-)authors is a member of the editorial board of *Atmospheric Chemistry and Physics*.

*Acknowledgements.* NM and JM would like to thank Jianhao Zhang and Graham Feingold for helpful discussions on an earlier version of the
analysis. NM would like to thank Wuyin Lin, Oksana Guba, Walter Hannah, Benjamin Hillman, Andrew Bradley, and Noel Keen for their
generosity, openness, and helpfulness in technical endeavors and knowledge sharing that paved the way for the underlying model runs. This
research was supported as part of the Energy Exascale Earth System Model (E3SM) project, funded by the U.S. Department of Energy, Office
of Science, Office of Biological and Environmental Research Earth Systems Model Development Program area of Earth and Environmental
System Modeling. The Pacific Northwest National Laboratory (PNNL) is operated for DOE by Battelle Memorial Institute under Contract no.
DE-AC06-76RLO1830. The data was produced using a high-performance computing cluster provided by the DOE Office of Science, Office
of Biological and Environmental Research Earth Systems Model Development Program area of Earth and Environmental System Modeling
program and operated by the Laboratory Computing Resource Center at Argonne National Laboratory. This research used resources of the
National Energy Research Scientific Computing Center (NERSC), a U.S. Department of Energy Office of Science User Facility located at
Lawrence Berkeley National Laboratory, operated under Contract No. DE-AC02-05CH11231 using NERSC award BER-ERCAP-0027116.

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
