# Peer review of "Present-day correlations insufficient to predict cloud albedo change by anthropogenic aerosols in E3SM v2"

_EGUsphere, 2024_

## Referee Comment (RC2)

This is a very interesting study, in a sense that it lets the reader take away with motivating-unresolved questions rather than answers. To briefly summarize the study, the authors apply an innovative diagnostic technique, that was recently developed to use on satellite observations, to a climate model (E3SMv2) in which they can turn off/down/up parameterized processes related to aerosol-cloud interactions (ACI). They conclude that spatial correlations between Ac and Nd from either clean or polluted climate states fail to predict a change in Ac from direct perturbed-control climate simulations (i.e., present-day versus pre-industrial).

After reading the manuscript, I take away three key points from this work:

- Present day Ac-Nd small-scale spatial correlation (regression slope) is completely opposite between E3SMv2 and satellite observations, which remains an open-question to be resolved.
- The process sensitivity experiments confirm that spatial regression method ("the innovative diagnostic technique" that has been used in satellite studies to infer cloud albedo susceptibility) is indeed providing process level understanding, as turning off/down/up sensitivity parameters in E3SMv2 does change spatial-regression derived susceptibilities in the direction that one would expect from physical understanding of ACI.
- Even in a "world" with no precipitation-suppression mechanism and quadrupled sedimentation-entrainment feedback, a polluted climate simulation still possesses brighter clouds than a cleaner climate simulation, a response that is in the opposite direction of our physical expectation. This remains an open-question as well.

The manuscript is very well written in a super concise and direct way. As a reviewer and someone who works in the direct field, I appreciate the clear and concise writing which made the reviewing process very efficient! That being said, from a general ACP reader perspective, I think some necessary contexts, descriptions, and discussion remain to be added.

I think this research is highly worthy of publication, but I do have some concerns and questions that I think the authors should address first.

**Major comments:**

- Regarding the sensitivity experiments on sedimentation (or scaling fall velocity), I am quite confused. First, are you turning off/up/down (or scaling) the sensitivity of droplet fall speed to drop size or the actual fall velocity for all drops? If latter, I don't think it makes sense as it affects all drops with different sizes in the same manner and your size-dependence is not affected, plus how would quadrupled fall speed for all drops and thereby sedimentation flux lead to enhanced size-dependence of sedimentation (the essence of sedimentation-entrainment feedback). If former, 4 times the sensitivity of sedimentation to drop size leads to enhanced entrainment driven darkening potential makes perfect sense to me. However, I suspect that's not the case as I was totally lost in the statements between lines 147-153.
- In general, although I appreciate the stimulating questions raised by this work, I feel that the authors should provide some of their interpretations and/or speculations at the very least in a *Discussion* section.

o Regarding the present-day Ac-Nd spatial correlation comparison between E3SMv2 and satellite, an "apple-to-apple" comparison in my view, I wonder if it is related to the representation of the stratocumulus deck in E3SMv2? Are these clouds precipitating under conditions we expect them to precipitate? Could you show the simulated maps of cloud field and probability of precipitation? What can possibly lead to the completely opposite susceptibility pattern in the LWP-Nd space (no matter causal or dur to confounding), this troubles me quite a lot, some speculations would help, I think.

o Regarding the result that Ac-Nd spatial correlation do not predict the PD-PI experiment, an "apple-to-orange" comparison in my view, first, when you say "…everything else being held constant" (lines 161-162), does it imply no circulation changes can be attributed to the brightening signal (by keeping the exact same meteorology all the time)? does it mean there is absolutely no other feedbacks (large-scale) contributing to this brightening? If so, is it possible that the cloud regime totally changes between PD and PI and you are comparing stratocumulus (in PD) to cumulus (in PI) (for example, perhaps?) I think showing the actual cloud field in PI and PD simulations would help a lot.

o The main conclusion relies on the assumption that the simulate PD-PI results represent the true aerosol effect, but without these above mentioned feedbacks, to what extent do you think this assumption is robust?

o I think in the *Discussion* section, the fundamental difference between climate simulations and satellite observations should be discussed, in a sense that what should we expect when we see a difference in Ac-Nd spatial correlations between observations and simulations. Also, to some extent, we expect the PD-PI simulation from GCMs to overestimate cloud brightening, compared to observations.

o Does this configuration of your model produce an inverted-V LWP-Nd mean-state relationship for the NE Pacific?

**Minor comments:**

▪ I do not quite get the (main conclusion) statement of "present-day correlations constrain cloud albedo change by anthropogenic aerosols." Essentially, you have one simulated albedo change that the present-day correlation failed to predict. So, I would recommend rephrase this statement.

▪ Line 56, what's vertical grid spacing in this configuration? Do you need to refine it to capture the observed stratocumulus field in this region?

▪ Lines 73-74, does this mean you have to force the winds and large-scale circulation to PD conditions and if so, does this creates an energy imbalance in the simulation?

▪ Line 81, even limited to only daytime, you would get some variations in susceptibility (as has been shown between Terra and Aqua observations), I don't think it will change the conclusion, but just curious about whether you see variations in susceptibility between morning and afternoon?

▪ Line 88, what is the purpose of this minimum insolation of 575 $Wm^{-2}$ threshold? Related to solar zenith angle?

▪ Figure 2, before I was about to post this report, I saw the reply made by the authors to the 1st reviewer, and I realized this figure has been updated.

- Lines 185-187, I think some contexts are needed for "cloud seeding proposals" "marine cloud brightening" for a general reader.
- All figures, perhaps roughly indicating an effective radius isoline similarly to Zhang et al. (2022) and Zhang & Feingold (2023) helps to discern the LWP-Nd region with high likelihood of precipitation?

---

## Author Comment (AC1)

| Description | Figure 2 |
|---|---|
| Wrong (current) |
[Figure]
 |
| Fixed (revised) | |

---

## Author Response (AR1)

**Re: egusphere-2024-366**

**April 2024**

We appreciate the Reviewers' time and effort in engaging this manuscript. Our response is provided in blue to the Reviewers' comments in black.

**1  Reviewer 1**

In this manuscript, the cloud albedo susceptibility to cloud droplet number concentration (CDNC) is analyzed in an Earth system model (E3SM). Following the method of Zhang et. al. (2022), the susceptibility is analyzed in LWP-CDNC state space where the brightening and darkening regimes can be attributed to the cloud adjustments that are likely in those regimes given the cloud state. Starting from a default simulation, using a case based in the NE Pacific, the regimes are not so clear for the model result here as compared with the satellite composites (Zhang et. al., 2022). The manuscript then investigates two cloud adjustments, precipitation suppression and the sedimentation–entrainment feedback, through process denial and then process scaling. In both cases, scaling the process up increases the intensity of the response to aerosol perturbation. One of these experiments, where the dependence of autoconversion on droplet number is removed and fall speeds are 4x higher, is then used to compare pre-industrial aerosol conditions with present day. In both cases the cloud albedo susceptibility is negative throughout the whole LWP–CDNC state space however the increase in CDNC from PI to PD still results in an increase in albedo and hence the key result.

This manuscript contains an interesting set of experiments that nicely build on the work of Zhang et. al. (2022) and Zhang and Feingold (2023), and it is written to a high standard. The figures are clearly presented and easy to understand given the captions and analysis explanation. However, although I appreciate the brevity of the manuscript, it comes at the expense of clarity in several places. As the reader, I had to work quite hard at times to follow the logic, which does appear sound but could benefit from helping the reader fill in the gaps. My main comment on the method itself is about the presentation of this as a constraint method, which I think requires some clarification on how the cloud albedo change would be constrained. My recommendation to the editor is that this manuscript is accepted subject to minor revisions surrounding the presentation of the work.

Many thanks and much appreciation to the Reviewer! We have edited the manuscript according to the feedback to improve the clarity and presentation.

**1.1  Minor comments**

1. The main finding of this manuscript is that the present-day correlations are insufficient to constrain pre-industrial albedo change due to aerosol perturbations, but as a reader I am struggling to understand this key point. The result shows that using the case of enhanced sedimentation-entrainment feedback produces a negative cloud albedo susceptibility in both PI and PD and as such one would expect that the increase in aerosol from PI to PD would decrease cloud albedo, which does not occur. This is shown and described clearly. However, supposing cloud albedo had done as expected, how would the result be used to constrain cloud albedo change due to aerosol perturbations? My understanding of "constrain" is reducing the uncertain range of some value through ruling out implausible regions. Here it seems to be used more to do with whether the model is representative of what we would expect. The finding that the model does not respond according to this susceptibility is interesting and significant, but it seems to be more about a systemic error.

   The Reviewer's point is well taken. The main conclusion is rephrased throughout the manuscript to better indicate predictiveness rather than constraining outstanding uncertainty. As an example, the title is changed: Present-day correlations insufficient to  *predict* cloud albedo change by anthropogenic aerosols in E3SM v2.

   *Action*: Rephrased the main conclusion throughout the manuscript where appropriate.

2. The sedimentation experiment is one of the points that I think would benefit from more explanation. For the precipitation suppression, what is expected to happen and why is clear. But for the sedimentation the reader could be assisted in understanding the expectation. The paragraph beginning at line 137 describes what is done and the effect seen and the paragraph beginning at line 144 seems like it is going to explain why that might be so. But it only says that increasing the fall speed is expected to increase the sedimentation–entrainment feedback, which has already been shown in the result. As the reader, I have had to sit and think about what I would expect to happen physically. I expect that in denying sedimentation, or lessening the fall speed, the droplets hang around longer in the cloud top region and therefore would cause a stronger darkening since they have more time to evaporate and cause the entrainment feedback. In fact, it seems to me that there is increased darkening in Figure 3, leftmost plot, but this is not mentioned in the discussion. Following the logic for lessening the fall speed,

I would have expected that increasing the fall speed would decrease the sedimentation–entrainment feedback since the droplets are removed from the cloud top region before significant evaporation can take place. I can see the result in the rightmost plot shows the opposite, in line with what the authors postulate. The authors could consider taking the reader with them on why they postulate this, even if definitively proving the mechanism is outside the scope of this study. Are there any readily available diagnostics that could help explain this?

> In the MG2 scheme, which is used in E3SM v2, cloud droplet sedimentation is parameterized following the empirical relationship of single particle fall speed of $v_{\text{sed}} = aD^b$ where $a$ and $b$ are two constants (for cloud water, $a = 3 \times 10^7$ m$^{1-b}$ s$^{-1}$ and $b = 2$). This $v_{\text{sed}}$ gets transformed to its number and mass fluxes, in which the expressions become dependent on the parameterized size distribution (and not explicitly on droplet size), and is described along with the parameters above on page 3647 of the 2008 MG paper (doi: 10.1175/2008JCLI2105.1).

> *Action*: Added the equation and reference for sedimentation formulation when it is first described in the Methods section.

> *Action*: Edited the paragraph on this specific topic to read: "In increasing the sedimentation fall speed by a factor of four, we set out to test whether or not we are able to enhance (and thus detect) the so-called sedimentation–entrainment feedback. In the original framing thereof, smaller droplets (resulting from a droplet number increase) evaporate faster and sediment less, increasing entrainment, and thus increasing the prevalence of smaller droplets — in a positive feedback loop (citing Bretherton et al. 2007 and Zhang et al. 2023). In Figure 3, we show that we are able to detect finger prints of the sedimentation–entrainment feedback, though we are not certain about the specific pathways. We speculate that the size-dependent formulation of droplet sedimentation is the key mechanism through which increasing sedimentation causes an increase in the sedimentation–entrainment feedback. By construction, $v_{\text{sed}}$ is proportional to the square of the droplet diameter (citing MG 2008 paper) and so when sedimentation is increased, there is a preferential sedimentation of larger droplets, leaving a distribution with more smaller droplets atop the cloud. We caution that definitively understanding the process(es) leading to the manifestation of this entrainment feedbacks in climate models is beyond the scope of this current manuscript. We note that recent work by (citing Mülmenstädt et al. 2024a, b)provides insights related to diagnosing entrainment in general circulation models."

3. Following on from the above, the brightening in the sedimentation denial experiment is presumably a strengthening of the Twomey effect. The authors might consider highlighting that this is most likely the Twomey effect and again suggesting why we see an increase in this effect related to the suppression of sedimentation. Perhaps because the smaller droplets are remaining at cloud top?

> Yes, in the absence of one mechanism (sedimentation–entrainment feedback) responsible for darkening, then the dominant brightening mechanism remaining is the Twomey effect, given that precipitation suppression is also turned off in that specific case.

> *Action*: Added the following sentence in first pointing out the trend of darkening/brightening due to the sedimentation studies: "In the leftmost panel, where both precipitation suppression and droplet sedimentation are turned off, it is possible to see the brightening Twomey effect taking hold without any competition for thinner, non-precipitating clouds (low LWP)."

4. In examining the precipitation suppression result, line 129–130 reads: "It is evident that precipitation suppression denial (leftmost panel) increases the prevalence and magnitude of cloud darkening due to increasing droplet number. Moreover, the precipitation suppression experiments show a gradual increase of cloud brightening as precipitation suppression is strengthened from left to right, culminating in mostly brightening clouds in the rightmost panel." I have spent some time looking at Figure 2 and I am struggling to see any difference between the first three plots. The authors could consider quantifying the brightening/darkening in some way to justify what is described in the text.

> The figure had a mistakenly repeated first panel three times. The figure has been corrected and is also attached below in Figure 1 in this response document.

> *Action*: Fixed the figure.

5. The introduction is well written and very concise, however as the reader I want a little bit more to motivate the study, the approach taken and to fill in some blanks.

 (a) The authors could consider adding a fuller description of the sedimentation–entrainment feedback in paragraph 2. Not all readers will know what this is, and since the rest of the paragraph has explained the other adjustments clearly it is a shame to leave this one as a vague point. I notice that it is described later when analyzing the results, but that could even be moved up to the introduction and then referred to later.

> *Action*: Added the following three sentence fo the paragraph 2: "Both entrainment feedbacks are positive resulting from smaller droplets atop the clouds. For the evaporation–entrainment feedback, smaller droplets evaporate faster, inducing more cooling and mixing, which in turn induces more droplet evaporation. For the sedimentation–entrainment

[Figure]

Figure 1: The new corrected figure of aerosol-induced precipitation suppression scaling experiments.

feedback, smaller droplets decrease sedimentation flux atop the clouds, thus increasing cooling, which in turn increases the entrainment rate."

(b) Also in paragraph 2, the adjustments start with "For thin, non-precipitating clouds" but does not go on to point out where thicker or precipitating clouds come into it. Perhaps the authors could consider adding something along the lines of "for thicker clouds likely to precipitate" somewhere in the lifetime effect.

*Action*: Added "For thicker clouds that are likely to precipitate" in front of of the sentence discussion the lifetime effect, and removed "yet" from that sentence.

(c) A description of the outcomes from Zhang et. al. (2022) and Zhang and Feingold (2023) does not appear until the first paragraph of the results section, but again, the authors could consider having an overview of their findings in the introduction. Especially because the paragraph beginning on line 25 talks about the different regimes and patterns they find, as the reader I feel left in the dark about what these are. This would follow on really nicely from the previous paragraph where these physical adjustments are described.

*Action*: Added the following sentence in describing their results: "In particular, their results reveal that the brightening Twomey effect is most dominant for thinner, non-precipitating clouds as well as the brightening cloud lifetime effect is most dominant for thicker clouds; on the other hand, the darkening cloud-thinning processes compete in regimes in between."

(d) The authors could consider adding some more references to the introduction, particularly in the first paragraph when discussing cloud adjustments (second sentence). They could also consider adding an extra sentence stating the current understanding of the cloud radiative forcing in that overall, it is negative.

*Action*: Added the word "negative" in front of "radiative forcing response to a cloud droplet perturbation" in the last sentence of the first paragraph.

6. The methods section is quite comprehensive with an extensive description of the setup of the simulations. The authors could consider including an equation to show how the fall speed is used in the sedimentation parameterization. The autoconversion is partly given, but they could also consider showing the full equation, including the dependence on specific humidity since it is a fairly simple equation.

*Action*: Added the fuller autoconversion equation as well as the droplet sedimentation details, with precise references to their sources.

**1.2 References**

- Zhang, J. and Feingold, G.: Distinct regional meteorological influences on low-cloud albedo susceptibility over global marine stratocumulus regions, Atmospheric Chemistry and Physics, 23, 1073–1090, https://doi.org/10.5194/acp-23-1073-2023, 2023

- Zhang, J., Zhou, X., Goren, T., and Feingold, G.: Albedo susceptibility of northeastern Pacific stratocumulus: the role of covarying meteorological conditions, Atmospheric Chemistry and Physics, 22, 861–880, https://doi.org/10.5194/acp-22-861-2022, 2022

**2 Reviewer 2**

This is a very interesting study, in a sense that it lets the reader take away with motivating- unresolved questions rather than answers. To briefly summarize the study, the authors apply an innovative diagnostic technique, that was recently developed to use on satellite observations, to a climate model (E3SMv2) in which they can turn off/down/up parameterized processes related to aerosol-cloud interactions (ACI). They conclude that spatial correlations between Ac and Nd from either clean or polluted climate states fail to predict a change in Ac from direct perturbed-control climate simulations (i.e., present-day versus pre-industrial).

After reading the manuscript, I take away three key points from this work:

- Present day Ac-Nd small-scale spatial correlation (regression slope) is completely opposite between E3SMv2 and satellite observations, which remains an open-question to be resolved.

- The process sensitivity experiments confirm that spatial regression method ("the innovative diagnostic technique" that has been used in satellite studies to infer cloud albedo susceptibility) is indeed providing process level understanding, as turning off/down/up sensitivity parameters in E3SMv2 does change spatial-regression derived susceptibilities in the direction that one would expect from physical understanding of ACI.

- Even in a "world" with no precipitation-suppression mechanism and quadrupled sedimentation-entrainment feedback, a polluted climate simulation still possesses brighter clouds than a cleaner climate simulation, a response that is in the opposite direction of our physical expectation. This remains an open-question as well.

The manuscript is very well written in a super concise and direct way. As a reviewer and someone who works in the direct field, I appreciate the clear and concise writing which made the reviewing process very efficient! That being said, from a general ACP reader perspective, I think some necessary contexts, descriptions, and discussion remain to be added.

I think this research is highly worthy of publication, but I do have some concerns and questions that I think the authors should address first.

> Many thanks and much appreciation to the Reviewer! Our sincere hope is that the general *ACP* reader will appreciate the concise and direct nature of this manuscript, much like the Reviewer did.

**2.1 Major comments**

- Regarding the sensitivity experiments on sedimentation (or scaling fall velocity), I am quite confused. First, are you turning off/up/down (or scaling) the sensitivity of droplet fall speed to drop size or the actual fall velocity for all drops? If latter, I don't think it makes sense as it affects all drops with different sizes in the same manner and your size-dependence is not affected, plus how would quadrupled fall speed for all drops and thereby sedimentation flux lead to enhanced size-dependence of sedimentation (the essence of sedimentation-entrainment feedback). If former, 4 times the sensitivity of sedimentation to drop size leads to enhanced entrainment driven darkening potential makes perfect sense to me. However, I suspect that's not the case as I was totally lost in the statements between lines 147-153.

> We scale the actual fall velocity of all droplets, but that still has a size dependence. We repeat our response to Reviewer 1 on this topic. In the MG2 scheme, which is used in E3SM v2, cloud droplet sedimentation is parameterized following the empirical relationship of single particle fall speed of $v_{\text{sed}} = aD^b$ where $a$ and $b$ are two constants (for cloud water, $a = 3 \times 10^7$ m$^{1-b}$ s$^{-1}$ and $b = 2$). This $v_{\text{sed}}$ gets transformed to its number and mass fluxes, in which the expressions become dependent on the parameterized size distribution (and not explicitly on droplet size), and is described along with the parameters above on page 3647 of the 2008 MG paper (doi: 10.1175/2008JCLI2105.1).
>
> *Action*: Added the equation and reference for sedimentation formulation when it is first described in the Methods section.
>
> *Action*: Edited the paragraph on this specific topic to read: "In increasing the sedimentation fall speed by a factor of four, we set out to test whether or not we are able to enhance (and thus detect) the so-called sedimentation–entrainment feedback. In the original framing thereof, smaller droplets (resulting from a droplet number increase) evaporate faster and sediment less, increasing entrainment, and thus increasing the prevalence of smaller droplets — in a positive feedback loop (citing Bretherton et al. 2007 and Zhang et al. 2023). In Figure 3, we show that we are able to detect finger prints of the sedimentation–entrainment feedback, though we are not certain about the specific pathways. We speculate that the size-dependent formulation of droplet sedimentation is the key mechanism through which increasing sedimentation causes an increase in the sedimentation–entrainment feedback. By construction, $v_{\text{sed}}$ is proportional to the square of the droplet diameter (citing MG 2008 paper) and so when sedimentation is increased, there is a preferential sedimentation of larger droplets, leaving a distribution with more smaller droplets atop the cloud. We caution that definitively understanding the process(es) leading to the manifestation of this entrainment feedbacks in climate models is beyond the scope of this current manuscript. We note that recent work by (citing Mülmenstädt et al. 2024a, b)provides insights related to diagnosing entrainment in general circulation models."

- In general, although I appreciate the stimulating questions raised by this work, I feel that the authors should provide some of their interpretations and/or speculations at the very least in a Discussion section.

> We have added more discussion within the Results section, instead of in a separate dedicated section. While we prefer to minimize speculation, we have added some additional interpretation and potential hypotheses in response to the Reviewers' comments. We finally point out that further interpretation and insight are discussed in related papers by Mülmenstädt et al. (2024a, b), which are now cited in this manuscript. Our hope for this specific manuscript is to provide a direct and concise result about what could be framed as a puzzle — that people in our field will be working hard to resolve in the near future in a productive scholarly conversation.

*Action*: Added a reference to Mülmenstädt et al. (2024a, b) in the main text at the end of the entrainment results: "We note that recent work by Mülmenstädt et al. (2024a, b) provides insights related to diagnosing entrainment in general circulation models."

– Regarding the present-day Ac-Nd spatial correlation comparison between E3SMv2 and satellite, an "apple-to-apple" comparison in my view, I wonder if it is related to the representation of the stratocumulus deck in E3SMv2? Are these clouds precipitating under conditions we expect them to precipitate? Could you show the simulated maps of cloud field and probability of precipitation? What can possibly lead to the completely opposite susceptibility pattern in the LWP-Nd space (no matter causal or dur to confounding), this troubles me quite a lot, some speculations would help, I think.

We would like to stress that it is not the goal of this work to compare satellite observations to model studies directly. For example, the variables studied in this work are diagnosed at cloud top with the maximum–random overlap algorithm, instead of satellite simulator outputs. So, in general, comparing these results against satellite observations in an apples-to-apples fashion may not be appropriate. The goal is to show that applying this diagnostic framework could be enlightening, but at the same time, it culminates in a puzzling null result. As for the simulated cloud fields, Figure 2 in this response document shows the total simulated cloud fields corresponding to Figure 4 in the submitted manuscript. It shows that the total cloud fields in the Northeast Pacific are largely consistent in PI and PD (slightly more cloudy — in total — in PD than PI). One potential explanation for the puzzling null result could be (further, confounding) meteorological covariability, and that will be the subject of future studies.

We additionally disagree with the Reviewer that E3SM v2 results are completely opposite to satellite observations. We believe the model results do capture the Twomey effect relatively well in Figure 1 in the submitted manuscript. The competition between the entrainment feedbacks and precipitation suppression seems to be more complex. The bottom line is, despite the null result, we were inspired by the Zhang et al. (2023) method, and their has proven very effective at capturing nuanced process-level details in the model. For example, both Figure 2 and Figure 3 in the submitted manuscript show remarkable consistency between what we expect the physics to do and what the diagnostic method is showing.

*Action*: Added a brief discussion with references to limitations and other challenges on the satellite–model front as well as the biases of E3SM v2: "We note that reconciling satellite and model studies is outside the scope of this work, and there exist significant challenges in that regard (citing e.g., Ma et al. 2018 and Quaas et al. 2020), and as such, we do not emphasize a direct comparison between our results and those of (citing Zhang et al. 2023). We additionally note the E3SM v2 model exhibits a "too-frequent, too-light" precipitation bias among other biases in simulated cloud fields as studied elsewhere (citing e.g., Xie et al. 2019 and Zhang et al. 2024), and we do not quantify them here."

*Action*: Added the following paragraph in the process sensitivity section to reiterate the value of the diagnostic method despite the imperfect alignment between model studies and observations: "Overall, our results in Figure 2 and Figure 3 show that diagnosing cloud albedo susceptibility in the CDNC–LWP variable space (citing Zhang et al. 2022, 2023) provides process-level understanding largely consistent with and explainable by our current understanding of physics involved. In Figure 2, manipulating autoconversion parameter to increase precipitation suppression does in fact result in more brightening cloud regimes. In Figure 3, manipulating the size-dependent cloud droplet sedimentation fall speed to increase the sedimentation–entrainment feedback results in more darkening regimes. Together, both Figure 2 and Figure 3 bolster our confidence in the valuable insight provided by this method and its ability to diagnose the underlying physics, despite the imperfect alignment between results obtained from satellite observations (citing Zhang et al. 2022, 2023) and model simulations (this work)."

– Regarding the result that Ac–Nd spatial correlation do not predict the PD–PI experiment, an "apple-to-orange" comparison in my view, first, when you say "…everything else being held constant" (lines 161-162), does it imply no circulation changes can be attributed to the brightening signal (by keeping the exact same meteorology all the time)? does it mean there is absolutely no other feedbacks (large-scale) contributing to this brightening? If so, is it possible that the cloud regime totally changes between PD and PI and you are comparing stratocumulus (in PD) to cumulus (in PI) (for example, perhaps?) I think showing the actual cloud field in PI and PD simulations would help a lot.

We believe it is not reasonable to argue that there are "absolutely no other feedbacks" in this case. The model has many interactive components, including a microphysics scheme that interact with a macrophysics scheme, and with a prognostic aerosol scheme as well. We are sampling at a three-hourly rate (at a diagnosed cloud top), which may or may not be enough to capture all the details in question. We are only looking at a select subset of variables. The cloud regime is not changing significantly between PD and PI, as evident in Figure 2 added to this response document, showing that the cloud fields in PD and PI are consistent with each other.

*Action*: Added a brief discussion with references to studies on biases of E3SM v2 when it comes to simulated cloud fields and precipitation: "We additionally note the E3SM v2 model exhibits a "too-frequent, too-light" precipitation bias among other biases in simulated cloud fields as studied elsewhere (citing e.g., Xie et al. 2019 and Zhang et al. 2024), and we do not quantify them here."

– The main conclusion relies on the assumption that the simulate PD–PI results represent the true aerosol effect, but without these above mentioned feedbacks, to what extent do you think this assumption is robust?

We do not necessarily think these PD–PI results represent the "true" aerosol effect, but they do represent the current standard protocol on how to quantify the aerosol effective radiative forcing in climate models, and that is applied to E3SM v2. So, as the Reviewer noted, we show that diagnosing cloud albedo susceptibility seems robust enough (as cursorily shown through the section on process studies in the submitted and revised manuscript). Then, when looking at what the present-day correlations "predict" the anthropogenic aerosol effect to be, we arrive at a null result when we conduct a pre-industrial simulation (or vice versa). The null result does not explicitly depend on the aerosol effect — the null result is simply that a diagnosed negative cloud albedo susceptibility (by definition, cloud albedo change divided by cloud droplet change) does not materialize in cloud albedo change measured between two experiments where the cloud droplet change is positive.

– I think in the Discussion section, the fundamental difference between climate simulations and satellite observations should be discussed, in a sense that what should we expect when we see a difference in Ac-Nd spatial correlations between observations and simulations. Also, to some extent, we expect the PD-PI simulation from GCMs to overestimate cloud brightening, compared to observations.

We agree the results could be an artifact of what we expect in terms of GCMs overestimating cloud brightening. As both Reviewers noted, the puzzling null result could due to a systemic error in the GCM at hand or something else — that remains an open question.

*Action*: Added a brief discussion with references to limitations and other challenges on the satellite–model comparison: "We note that reconciling satellite and model studies is outside the scope of this work, and there exist significant challenges in that regard (citing e.g., Ma et al. 2018 and Quaas et al. 2020), and as such, we do not emphasize a direct comparison between our results and those of (citing Zhang et al. 2023)."

– Does this configuration of your model produce an inverted-V LWP–Nd mean-state relationship for the NE Pacific?

Yes, this configuration shares much of what Mülmenstädt et al. (2024a, b) discuss in their manuscripts, including the inverted-V. See Figure 3.

**2.2 Minor comments**

- I do not quite get the (main conclusion) statement of "present-day correlations constrain cloud albedo change by anthropogenic aerosols." Essentially, you have one simulated albedo change that the present-day correlation failed to predict. So, I would recommend rephrase this statement.

As we responded to Reviewer 1, the Reviewer's point is well taken. The main conclusion is rephrased throughout the manuscript to better indicate predictiveness rather than constraining outstanding uncertainty. As an example, the title is changed: Present-day correlations insufficient to *predict* cloud albedo change by anthropogenic aerosols in E3SM v2.

*Action*: Rephrased the main conclusion throughout the manuscript where appropriate.

- Line 56, what's vertical grid spacing in this configuration? Do you need to refine it to capture the observed stratocumulus field in this region?

In E3SM v1, the vertical spacing was increased from 30 layers (like was done in CAM5) to 72 layers, with most of the increase due to increased vertical resolution in the planetary boundary layer, which is expected to help with the simulation of low clouds. The E3SM v2 retains the same exact 72 vertical atmospheric layers, with spacing illustrated — very roughly — in Figure 4 in this response document. And no, the model studies in this manuscript did not deviate from the standard configuration regarding vertical resolution.

*Action*: Qualified "comparison" to indicate the context of resolution by adding "in terms of resolution" after it.

- Lines 73–74, does this mean you have to force the winds and large-scale circulation to PD conditions and if so, does this creates an energy imbalance in the simulation?

The specific lines in this comment (73–74) are meant to cover classic "forcers" and not the circulation aspects. The circulation aspects are weakly controlled by nudging, and yes, both present-day and pre-industrial cases are nudged to present-day meteorology. We note that nudging does create an energy imbalance.

*Action*: Replaced "conditions" with "settings" to avoid confusion.

- Line 81, even limited to only daytime, you would get some variations in susceptibility (as has been shown between Terra and Aqua observations), I don't think it will change the conclusion, but just curious about whether you see variations in susceptibility between morning and afternoon?

  We agree it will not change the conclusion. We include *any* snapshot meeting the criteria in the text (daytime, etc.) because we do not have an explicit Terra and Aqua simulator and because we want to have enough statistics (if we limited to one snapshot per day, we would have one eights of the eight years used analyzed by Zhang et al. 2023). In a future work, we are planning to investigate the temporal aspect of the analysis more thoroughly (including longer time periods and looking at specific snapshots in morning and afternoon).

  *Action*: Added a brief discussion with references to limitations and other challenges on the satellite–model comparison: "We note that reconciling satellite and model studies is outside the scope of this work, and there exist significant challenges in that regard (citing e.g., Ma et al. 2018 and Quaas et al. 2020), and as such, we do not emphasize a direct comparison between our results and those of (citing Zhang et al. 2023)."

- Line 88, what is the purpose of this minimum insolation of 575 W m$^{-2}$ threshold? Related to solar zenith angle?

  Yes, largely to follow the Zhang et al. (2022) filtering.

  *Action*: Rephrased to make it clear the context of this specific number: "In particular, we select points satisfying solar zenith angle less than 65 ° or minimum solar insolation of 575 W m$^{-2}$ …".

- Figure 2, before I was about to post this report, I saw the reply made by the authors to the 1st reviewer, and I realized this figure has been updated.

  Like I responded to Reviewer 1, I apologize for the confusion resulting from my mistake in producing this specific figure. In attempting to edit the figures, I mistakenly repeated the first panel three times. The figure has been corrected and is also attached below. I think the description in both the figure caption and the text stand. See Figure 1 in this response document.

  *Action*: Fixed the figure.

- Lines 185–187, I think some contexts are needed for "cloud seeding proposals" "marine cloud brightening" for a general reader.

  *Action*: Removed cloud seeding proposals from the penultimate sentence, and edited the last sentence to briefly explain the context: "While we focus on the present-day versus pre-industrial comparison in this study, we do not explicitly assess the implications for future scenarios, for example, for the purpose of better assessing cloud seeding proposals (e.g., marine cloud brightening) where the goal is to exploit cloud physics properties like cloud albedo susceptibility to cool the planet via a positive perturbation of cloud droplets via aerosols."

- All figures, perhaps roughly indicating an effective radius isoline similarly to Zhang et al. (2022) and Zhang & Feingold (2023) helps to discern the LWP-Nd region with high likelihood of precipitation?

  We agree that it *could* help. However, since this specific variable is not diagnosed in the model runs studied in the manuscript, we hesitate to superimpose it as it could be misleading with respect to the model's behavior. Nevertheless, Figure 5 in this response document shows the mean precipitation the same plot (with the same exact assumptions) as Figure 1 in the manuscript (it was produced by swapping cloud albedo susceptibility with precipitation). It shows that the pattern is generally consistent with that of Zhang et al. 2023, such that the effective radius isoline would be near the middle of the color map (white-ish).

[Figure]

Figure 2: Simulated total cloud fields averaged over the entire studied year (2011) in the Northeast Pacific corresponding to Figure 4 in the submitted manuscript as well as the difference between.

[Figure]

Figure 3: The inverted-V LWP–CDNC relationship in the default configuration corresponding to Figure 1 in the submitted manuscript.

[Figure]

Figure 4: The vertical profile in E3SM v2 diagnosed *very roughly* in an averaged January snapshot via the geopotential height and approximate pressure for illustration purposes only.

[Figure]

Figure 5: Large-scale precipitation (i.e., non-convective) in the default configuration corresponding to Figure 1 in the submitted manuscript.